# HIGHER-ORDER FUNCTION NETWORKS FOR LEARNING COMPOSABLE 3D OBJECT REPRESENTATIONS

**Eric Mitchell**[1][2]      **Selim Engin**[3][†]      **Volkan Isler**[2]      **Daniel D Lee**[2]

## ABSTRACT

We present a new approach to 3D object representation where a neural network encodes the geometry of an object directly into the weights and biases of a second 'mapping' network. This mapping network can be used to reconstruct an object by applying its encoded transformation to points randomly sampled from a simple geometric space, such as the unit sphere. We study the effectiveness of our method through various experiments on subsets of the ShapeNet dataset. We find that the proposed approach can reconstruct encoded objects with accuracy equal to or exceeding state-of-the-art methods with orders of magnitude fewer parameters. Our smallest mapping network has only about 7000 parameters and shows reconstruction quality on par with state-of-the-art object decoder architectures with millions of parameters. Further experiments on feature mixing through the composition of learned functions show that the encoding captures a meaningful subspace of objects. [‡]

## 1 INTRODUCTION

This paper is primarily concerned with the problem of learning compact 3D object representations and estimating them from images. If we consider an object to be a continuous surface in $\mathbb{R}^3$, it is not straightforward to directly represent this infinite set of points in memory. In working around this problem, many learning-based approaches to 3D object representation suffer from problems related to memory usage, computational burden, or sampling efficiency. Nonetheless, neural networks with tens of millions of parameters have proven effective tools for learning expressive representations of geometric data. In this work, we show that object geometries can be encoded into neural networks with thousands, rather than millions, of parameters with little or no loss in reconstruction quality.

To this end, we propose an object representation that encodes an object as a function that maps points from a canonical space, such as the unit sphere, to the set of points defining the object. In this work, the function is approximated with a small multilayer perceptron. The parameters of this function are estimated by a 'higher order' encoder network, thus motivating the name for our method: *Higher-Order Function networks (HOF)*. This procedure is shown in Figure 1. There are two key ideas that distinguish HOF from prior work in 3D object representation learning: fast-weights object encoding and interpolation through function composition.

*(1) Fast-weights object encoding:* 'Fast-weights' in this context generally refers to methods that use network weights and biases that are not fixed; at least some of these parameters are estimated on a per-sample basis. Our fast-weights approach stands in contrast to existing methods which encode objects as vector-valued inputs to a decoder network with fixed weights. Empirically, we find that our approach enables a dramatic reduction (two orders of magnitude) in the size of the mapping network compared to the decoder networks employed by other methods.

*(2) Interpolation through function composition:* Our functional formulation allows for interpolation between inputs by composing the roots of our reconstruction functions. We demonstrate that the

[1]Stanford University    [2]Samsung AI Center - New York    [3]University of Minnesota    [†]Work performed while an intern at Samsung AI Center - New York. [‡] See https://saic-ny.github.io/hof for code and additional information. Correspondence to: Eric Mitchell <eric.mitchell@cs.stanford.edu>.

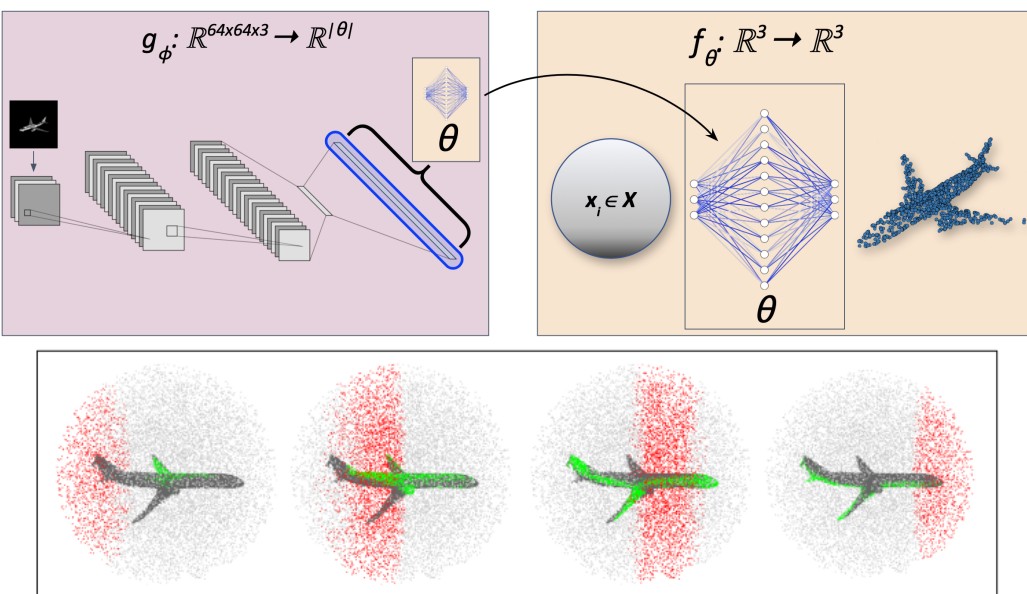

Figure 1: **Top**: Overview of HOF. The encoder network $g_\phi$ encodes the geometry of the object pictured in each input image directly into the parameters of the mapping function $f_\theta$, which produces a reconstruction as a transformation of a canonical object (here, the unit sphere). **Bottom**: We visualize the transformation $f_\theta$ by showing various subsets of the inputs $X$ and their corresponding mapped locations in red and green, respectively. In each frame, light gray shows the rest of $X$ and dark gray shows the rest of the reconstructed object.

functional representation learned by HOF provides a rich latent space in which we can 'interpolate' between objects, producing new, coherent objects sharing properties of the 'parent' objects.

In order to position HOF among other methods for 3D reconstruction, we first define a taxonomy of existing work and show that HOF provides a generalization of current best-performing methods. Afterwards, we demonstrate the effectiveness of HOF on the task of 3D reconstruction from an RGB image using a subset of the ShapeNet dataset (Chang et al., 2015). The results, reported in Tables 1 and 2 and Figure 2, show state-of-the-art reconstruction quality using orders of magnitude fewer parameters than other methods.

## 2 RELATED WORK

The selection of object representation is a crucial design choice for methods addressing 3D reconstruction. Voxel-based approaches (Choy et al., 2016; Häne et al., 2017) typically use a uniform discretization of $\mathbb{R}^3$ in order to extend highly successful convolutional neural network (CNN) based approaches to three dimensions. However, the inherent sparsity of surfaces in 3D space make voxelization inefficient in terms of both memory and computation time. Partition-based approaches such as octrees (Tatarchenko et al., 2017; Riegler et al., 2017) address the space efficiency shortcomings of voxelization, but they are tedious to implement and more computationally demanding to query. Graph-based models such as meshes (Wang et al., 2018; Gkioxari et al., 2019; Smith et al., 2019; Hanocka et al., 2019) provide a compact representation for capturing topology and surface level information, however their irregular structure makes them harder to learn. Point set representations, discrete (and typically finite) subsets of the continuous geometric object, have also gained popularity due to the fact that they retain the simplicity of voxel based methods while eliminating their storage and computational burden (Qi et al., 2017a; Fan et al., 2017; Qi et al., 2017b; Yang et al., 2018; Park et al., 2019). The PointNet architecture (Qi et al., 2017a;b) was an architectural milestone that made manipulating point sets with deep learning methods a competitive alternative to earlier approaches; however, PointNet is concerned with *processing*, rather than *generating*, point clouds. Further, while point clouds are more flexible than voxels in terms of information density, it is still not obvious how

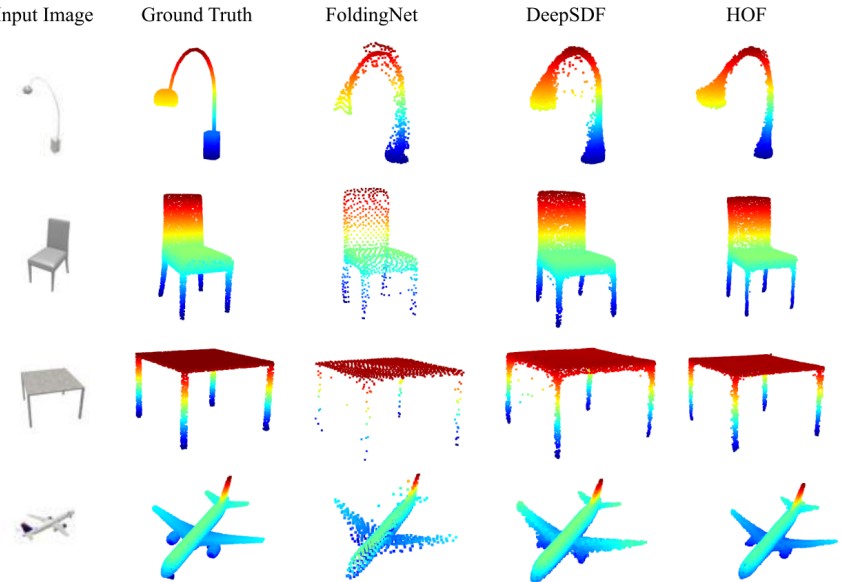

Figure 2: From left to right: Input RGB image, ground truth point cloud, reconstruction from FoldingNet (Yang et al., 2018), reconstruction from DeepSDF (Park et al., 2019), and our method.

to adapt them to the task of producing arbitrary- or varied-resolution predictions. Independently regressing each point in the point set requires additional parameters for each additional point (Fan et al., 2017; Achlioptas et al., 2018), which is an undesirable property if the goal is high-resolution point clouds.

Many current approaches to representation and reconstruction follow an encoder-decoder paradigm, where the encoder and decoder both have learned weights that are fixed at the end of training. An image or set of 3D points is encoded as a latent vector 'codeword' either with a learned encoder as in Yang et al. (2018); Lin et al. (2018); Yan et al. (2016) or by direct optimization of the latent vector itself with respect to a reconstruction-based objective function as in Park et al. (2019). Afterwards, the latent code is decoded by a learned decoder into a reconstruction of the desired object by one of two methods, which we call *direct decoding* and *contextual mapping*. Direct decoding methods directly map the latent code into a fixed set of points (Choy et al., 2016; Fan et al., 2017; Lin et al., 2018; Michalkiewicz et al., 2019); contextual mapping methods map the latent code into a function that can be sampled or otherwise manipulated to acquire a reconstruction (Yang et al., 2018; Park et al., 2019; Michalkiewicz et al., 2019; Mescheder et al., 2019). Direct decoding methods generally suffer from the limitation that their predictions are of fixed resolution; they cannot be sampled more or less precisely. With contextual mapping methods, it is possible in principle to sample the object to arbitrarily high resolution with the correct decoder function. However, sampling can provide a significant computational burden for some contextual mapping approaches as those proposed by Park et al. (2019) and Michalkiewicz et al. (2019). Another hurdle is the need for post-processing such as applying the Marching Cubes algorithm developed by Lorensen and Cline (1987). We call contextual mapping approaches that encode context by concatenating a duplicate of a latent context vector with each input *latent vector concatenation (LVC)* methods. In particular, we compare with LVC architectures used in FoldingNet (Yang et al., 2018) and DeepSDF (Park et al., 2019).

HOF is a contextual mapping method that distinguishes itself from other methods within this class through its approach to representing the mapping function: HOF uses one neural network to estimate the weights of another. Conceptually related methods have been previously studied under nomenclature such as the 'fast-weight' paradigm (Schmidhuber, 1992; De Brabandere et al., 2016; Klein et al., 2015; Riegler et al., 2015) and more recently 'hypernetworks' (Ha et al., 2016). However, the work by Schmidhuber (1992) deals with encoding memories in sequence learning tasks. Ha et al. (2016) suggest that estimating weights of one network with another might lead to improvements in parameter-efficiency. However, this work does not leverage the key insight of using network parameters that are estimated *per sample* in vision tasks.

## 3 HIGHER-ORDER FUNCTION NETWORKS

HOF is motivated by the independent observations by both Yang et al. (2018) and Park et al. (2019) that LVC methods do not perform competitively when the context vector is injected by simply concatenating it with each input. In both works, the LVC methods proposed required architectural workarounds to produce sufficient performance on reconstruction tasks, including injecting the latent code multiple times at various layers in the network. HOF does not suffer from these shortcomings due to its richer context encoding (the entire mapping network encodes context) in comparison with LVC. We compare the HOF and LVC regimes more precisely in Section 3.2. Quantitative comparisons of HOF with existing methods can be found in Table 1.

### 3.1 A FAST-WEIGHTS APPROACH TO 3D OBJECT REPRESENTATION AND RECONSTRUCTION

We consider the task of reconstructing an object point cloud $O$ from an image. We start by training a neural network $g_\phi$ with parameters $\phi$ (Figure 1, top-left) to output the parameters $\theta$ of a mapping function $f_\theta$, which reconstructs the object when applied to a set of points $X$ sampled uniformly from a canonical set such as the unit sphere (Figure 1, top-right). We note that the number of samples in $X$ can be increased or decreased to produce higher or lower resolution reconstructions without changing the network architecture or retraining, in contrast with direct decoding methods and some contextual mapping methods which use fixed, non-random samples from $X$ (Yang et al., 2018). The input to $g_\phi$ is an RGB image $I$; our implementation takes $64 \times 64 \times 3$ RGB images as input, but our method is general to any input representation for which a corresponding differentiable encoder network can be constructed to estimate $\theta$ (e.g. PointNet (Qi et al., 2017a) for point cloud completion). Given $I$, we compute the parameters of the mapping network $\theta_I$ as

$$\theta_I = g_\phi(I) \tag{1}$$

That is, the encoder $g_\phi : \mathbb{R}^{3 \times 64 \times 64} \to \mathbb{R}^d$ directly regresses the $d$-dimensional parameters $\theta_I$ of the mapping network $f_{\theta_I} : \mathbb{R}^c \to \mathbb{R}^3$, which maps $c$-dimensional points in the canonical space $X$ to points in the reconstruction $\hat{O}$ (see Figure 1). We then transform our canonical space $X$ with $f_{\theta_I}$ in the same manner as other contextual mapping methods:

$$\hat{O} = \{f_{\theta_I}(\mathbf{x_i}) : \mathbf{x_i} \in X\} \tag{2}$$

During training, we sample an image $I$ and the corresponding ground truth point cloud model $O$, where $O$ contains 10,000 points sampled from the surface of the true object. We then obtain the mapping $f_{\theta_I} = g_\phi(I)$ and produce an estimated reconstruction of $O$ as in Equation 2. In our training, we only compute $f_{\theta_I}(\mathbf{x})$ for a sample of 1000 points in $X$. However, we find that sampling many more points (10-100$\times$ as many) at test time still yields high-quality reconstructions. This sample is drawn from a uniform distribution over the set $X$. We then compute a loss for the prediction $\hat{O}$ using a differentiable set similarity metric such as Chamfer distance or Earth Mover's Distance. We focus on the Chamfer distance as both a training objective and metric for assessing reconstruction quality. The asymmetric Chamfer distance $CD(X, Y)$ is often used for quantifying the similarity of two point sets $X$ and $Y$ and is given as

$$CD(X, Y) = \frac{1}{|X|} \sum_{\mathbf{x}_i \in X} \min_{\mathbf{y}_i \in Y} ||\mathbf{x}_i - \mathbf{y}_i||_2^2 \tag{3}$$

The Chamfer distance is defined even if sets $X$ and $Y$ have different cardinality. We train $g_\phi$ to minimize the *symmetric* objective function $\ell(\hat{O}, O) = CD(\hat{O}, O) + CD(O, \hat{O})$ as in Fan et al. (2017).

### 3.2 COMPARISON WITH LVC METHODS

We compare our mapping approach with LVC architectures such as DeepSDF (Park et al., 2019) and FoldingNet (Yang et al., 2018). These architectures control the output of the decoder through the concatenation of a latent 'codeword' vector $\mathbf{z}$ with each input $\mathbf{x_i} \in X$. The codeword is estimated by an encoder $g_{\phi_{LVC}}$ for each image. We consider the case in which the latent vector is only concatenated

with inputs in the first layer of the decoder network $f_\theta$, which we assume to be an MLP. We are interested in analyzing the manner in which the network output with respect to $\mathbf{x_i}$ may be modulated by varying $\mathbf{z}$.

If the vector $\mathbf{a_i}$ contains the pre-activations of the first layer of $f_\theta$ given an input point $\mathbf{x_i}$, we have

$$\mathbf{a_i} = W^{\mathbf{x}}\mathbf{x_i} + W^{\mathbf{z}}\mathbf{z} + \mathbf{b}$$

where $W^{\mathbf{x}}$, $W^{\mathbf{z}}$, and $\mathbf{b}$ are fixed parameters of the decoder, and only $\mathbf{z}$ is a function of $I$. If we absorb the parameters $W^{\mathbf{z}}$ and $\mathbf{b}$ into the encoder parameters $\phi_{\text{LVC}}$ (as $W^{\mathbf{z}}$ and $\mathbf{b}$ are fixed for all $\mathbf{x_i}$), we can define a new, equivalent latent representation $\mathbf{b}^* = W^{\mathbf{z}}\mathbf{z} + \mathbf{b} = W^{\mathbf{z}}g_{\phi_{\text{LVC}}}(I) + \mathbf{b}$ and a new encoder function $h$ with parameters $\phi_{\text{LVC}} \cup \{W^{\mathbf{z}}, \mathbf{b}\}$ such that $h(I) = \mathbf{b}^*$. This gives

$$\mathbf{a_i} = W^{\mathbf{x}}\mathbf{x_i} + h(I)$$

Thus the LVC approach is equivalent to estimating a *fixed subset* of the parameters $\theta$ of the decoder $f_\theta$ on a per-sample basis (the first layer bias). From this perspective, HOF is an intuitive generalization: rather than estimating just the first layer bias, we allow our encoder to modulate all of the parameters in the decoder $f_\theta$ on a per-sample basis.

Having demonstrated HOF as a generalization of existing contextual mapping methods, in the next section, we present a novel application of contextual mapping that leverages the compositionality of the estimated mapping functions to aggregate features of multiple objects or multiple viewpoints of the same object.

### 3.3 EXTENDING CONTEXTUAL MAPPING METHODS: FEATURE AGGREGATION THROUGH FUNCTION COMPOSITION

An advantageous property of methods that use a latent codeword is that they have been empirically shown to learn a meaningful space of object geometries, in which interpolating between object encodings gives new, coherent object encodings (Fan et al., 2017; Yang et al., 2018; Park et al., 2019; Groueix et al., 2018). HOF, on the other hand, does not obviously share this property: interpolating between the mapping function parameters estimated for two different objects need not yield a new, coherent object as the prior work has shown that the solution space of 'good' neural networks is highly non-convex (Li et al., 2018). We demonstrate empirically in Figure 7 that naively interpolating between reconstruction function in the HOF regime does indeed produce meaningless blobs. However, with a small modification to the HOF formulation in Equation 2, we can in fact learn a rich space of *functions* in which we can interpolate between objects through function composition.

We extend the formulation in Equation 2 to one where an object is represented as the $k$-th power of the mapping $f_{\theta_I}$:

$$\hat{O} = \{f_{\theta_I}^k(\mathbf{x}) : \mathbf{x} \in X\} \tag{4}$$

where $f^k$ is defined as the composition of $f$ with itself $(k-1)$ times: $f^k(\mathbf{x}) = f(f^{(k-1)}(\mathbf{x}))$ where $f^0(\mathbf{x}) \triangleq \mathbf{x}$. We call a mapping $f_{\theta_I}$ whose $k$-th power reconstructs the object $O$ in image $I$ the $k$-mapping for $O$.

This modification to Equation 2 adds an additional constraint to the mapping: the domain and codomain must be the same. However, evaluating powers of $f$ leverages the power of weight sharing in neural network architectures; for an MLP mapping architecture with $l$ layers (excluding the input layer), evaluating its $k$-th power is equivalent to an MLP with $l \times k$ layers with shared weights. This formulation also has connections to earlier work on continuous attractor networks as a model for encoding memories in the brain as $k$ becomes large (Seung, 1998).

In Section 4.3, we conduct experiments in a setting in which we have acquired RGB images $I$ and $J$ of two objects, $O_I$ and $O_J$, respectively. Applying our encoder to these images, we obtain $k$-mappings $f_{\theta_I}$ and $f_{\theta_J}$, which have parameters $\theta_I = g_\phi(I)$ and $\theta_J = g_\phi(J)$, respectively. We hypothesize that we can combine the information contained in each mapping function $f_{\theta_i}$ by evaluating any of the $2^k$ possible functions of the form:

$$f_{\text{interp}} = (f_{\theta_1} \circ ... \circ f_{\theta_k}) \tag{5}$$

where the parameters of each mapping $f_{\theta_i}$ are either the parameters of $f_{\theta_I}$ or $f_{\theta_J}$. Figures 4 and 7 show that interpolation with function composition provides interesting, meaningful outputs in experiments with $k = 2$ and $k = 4$.

Table 1: Comparing various reconstruction architectures. Reported Chamfer distance values are multiplied by 100 for readability and include standard error in parentheses. HOF-1 and HOF-3 are HOF variants with 1 and 3 hidden layers, respectively.

| Method | CD(P,T) | CD(T,P) | Average CD | Parameters | Layers |
|--------|---------|---------|------------|------------|--------|
| 3D-R2N2 | 2.588 | 3.342 | 2.965 | 2,000,000+ | 12 |
| PSG | 1.982 | 2.146 | 2.064 | — | — |
| EPCG | 1.846 | 1.701 | 1.774 | 60,000,000+ | 8 |
| FoldingNet | 1.563 (0.0064) | 1.534 (0.0053) | 1.549 (0.0117) | 1,056,775 | 6 |
| DeepSDF | 1.521 (0.0029) | **0.962** (0.0018) | **1.242** (0.0047) | 1,578,241 | 8 |
| HOF-1 | 1.517 (0.0040) | 1.021 (0.0028) | 1.269 (0.0068) | **7,171** | **2** |
| HOF-3 | **1.480** (0.0028) | 1.014 (0.0020) | **1.247** (0.0048) | 34,052 | 4 |

## 4 EXPERIMENTAL EVALUATIONS

We conduct various empirical studies in order to justify two key claims. In Sections 4.1 and 4.2, we compare with other contextual mapping architectures to demonstrate that HOF provides equal or better performance with a significant reduction in parameters and compute time. In Section 4.3 we demonstrate that extending contextual mapping approaches such as HOF with multiple compositions of the mapping function provides a simple and effective approach to object interpolation. Further experimentation, including ablation studies and a simulated robot navigation scenario, can be found in A.7.

### 4.1 EVALUATING RECONSTRUCTION QUALITY ON SHAPENET

We test HOF's ability to reconstruct a 3D point cloud of an object given a single RGB image, comparing with other architectures for 3D reconstruction. We conduct two experiments:

1. We compare HOF with LVC architectures to show that HOF is a more parameter-efficient approach to contextual mapping than existing fixed-decoder architectures.

2. We compare HOF to a broader set of state of the art methods on a larger subset of the ShapeNet dataset, demonstrating that it matches or surpasses existing state of the art methods in terms of reconstruction quality.

In the first experiment, we compare HOF with other LVC architectures on 13 of the largest classes of ShapeNet (Yan et al., 2016), using the asymmetric Chamfer distance metrics (Equation 3) as reported in Lin et al. (2018). In the second experiment, we compare HOF with other methods for 3D reconstruction on a broader selection of 55 classes the ShapeNet dataset, as in Tatarchenko et al. (2019). In line with the recommendations in Tatarchenko et al. (2019), we report F1 scores for this evaluation.

### 4.1.1 LVC ARCHITECTURE COMPARISON

For this experiment, we compare HOF with LVC decoder architectures proposed in the literature, specifically those used in DeepSDF (Park et al., 2019) and FoldingNet (Yang et al., 2018), as well as several other baselines. Each architecture maps points from $\mathbb{R}^c$ to $\mathbb{R}^3$ in order to enable a direct comparison. The dataset contains 31773 ground truth point cloud models for training/validation and 7926 for testing. For each point cloud, there are 24 RGB renderings of the object from a fixed set of 24 camera positions. For both training and testing, each point cloud is shifted so that its bounding box center is at the origin in line with Fan et al. (2017). At test time, there is *no post-processing performed on the predicted point cloud*. The architectures we compare in this experiment are:

1. HOF-1: 1 hidden layer containing 1024 hidden neurons

2. HOF-3: 3 hidden layers containing 128 hidden neurons

3. DeepSDF as described in Park et al. (2019), with 8 hidden layers containing 512 neurons each

4. FoldingNet as described in Yang et al. (2018), with 2 successive 'folds', each with a 3-layer MLP with 512 hidden neurons

5. EPCG architecture as reported in Lin et al. (2018)

6. Point Set Generation network (Fan et al., 2017) as reported in Lin et al. (2018)

7. 3D-R2N2 (Choy et al., 2016) as reported in Lin et al. (2018)

Results are reported in Table 1. Chamfer Distance scores are scaled by 100 as in line with Lin et al. (2018). We find that HOF performs significantly better than that direct decoding baseline of Lin et al. (2018) and performs on par with other contextual mapping approaches with $30\times$ fewer parameters. In order to provide a fair comparison with the baseline method, we ensure that ground truth objects are scaled identically to those in Lin et al. (2018). We report both 'forward' Chamfer distance CD(Pred, Target) and 'backward' Chamfer distance CD(Target, Pred), again in line with the convention established by Lin et al. (2018). Table 7 contains a class-wise breakdown. Qualitative comparisons of the outputs of HOF with state-of-the-art architectures are shown in Figure 2.

### 4.1.2 SHAPENET BREADTH COMPARISON

Tatarchenko et al. (2019) question the common practice in single-view 3D reconstruction of evaluating only on the largest classes of ShapeNet. The authors demonstrate that reconstruction methods do not exhibit performance correlated with the size of object classes, and thus evaluating on smaller ShapeNet classes is justified. We use the dataset provided by Tatarchenko et al. (2019), which includes 55 classes from the ShapeNet dataset. The authors also suggest using the F1 score metric, defined as the harmonic mean between precision and recall (Tatarchenko et al., 2019).

Table 2: F1/class-weighted F1 score comparison of HOF with methods reported in Tatarchenko et al. (2019). Higher is better.

| Method | F1 | cw-F1 |
|---|---|---|
| *Oracle NN* | *0.290* | *0.321* |
| AtlasNet | 0.252 | 0.287 |
| OGN | 0.217 | 0.230 |
| Matryoshka | 0.264 | 0.297 |
| Retrieval | 0.237 | 0.260 |
| HOF (Ours) | **0.291** | **0.310** |

We include comparisons with AtlasNet (Groueix et al., 2018), Octree Generating Networks (Tatarchenko et al., 2017), Matryoshka Networks (Richter and Roth, 2018), and retrieval baselines as reported by Tatarchenko et al. (2019). We find that HOF performs competitively with all of these state-of-the-art methods, even surpassing them on many classes. We show summary statistics in Table 2. The F1 column contains the average F1 score for each method, uniformly averaging over all classes regardless of how class imbalances in the testing set. The cw-F1 score column averages over class F1 scores weighted by the fraction of the dataset comprised by that class; that is, classes that are over-represented in the testing set are correspondingly over-represented in the cw-F1 score. On the mean F1 metric, HOF outperforms all other methods, including the 'Oracle Nearest-Neighbor' approach described by Tatarchenko et al. (2019). The Oracle Nearest Neighbor uses the closest object in the training set as its prediction for each test sample. A complete class-wise performance breakdown is in the Appendix in Table 8. We find that HOF outperforms all 5 comparison methods (including the Oracle) in 23 of the 55 classes. Excluding the oracle, HOF shows the best performance in 28 of the 55 classes.

### 4.2 RUNTIME PERFORMANCE COMPARISON

We compare HOF with the decoder architectures proposed in Park et al. (2019) and Yang et al. (2018) in terms of inference speed. Figure 3 shows the results of this experiment, comparing how long it takes for each network to map a set of $N$ samples from the canonical space $X$ into the object reconstruction. We ignore the processing time for estimating the latent state $\mathbf{z}$ for DeepSDF/FoldingNet and the function parameters $\theta$ for HOF; we use the same convolutional neural network architecture with a modified output layer for both. We find that even for medium-resolution reconstructions (N > 1000), the GPU running times for the DeepSDF/FoldingNet architectures and HOF begin to diverge. This difference is even more noticeable in the CPU running time comparison (an almost $100\times$ difference). This performance improvement may be significant for embedded systems that need to efficiently

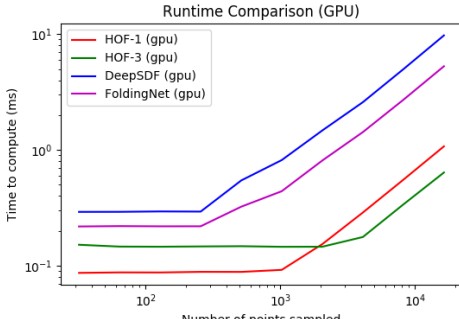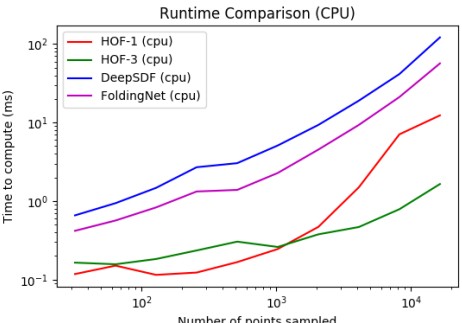

Figure 3: Runtime analysis comparing HOF with DeepSDF and FoldingNet architectures. HOF-1 and HOF-3 are HOF with 1 and 3 hidden layers, respectively. Computing environment details are given in Section B.2.

store and reconstruct 3D objects in real time; our representation is small in size, straightforward to sample uniformly (unlike a CAD model), and fast to evaluate.

### 4.3 OBJECT INTERPOLATION

To demonstrate that our functional representation yields an expressive latent object space, we show that the composition of these functions produces interesting, new objects. The top of Figure 4 shows in detail the composition procedure. If we have estimated 2-mappings for two objects $O_A$ and $O_B$, we demonstrate that $f_{\theta_A}(f_{\theta_B}(X))$ and $f_{\theta_B}(f_{\theta_A}(X))$ both provide interesting mixtures of the two objects and mix the features of the objects in different ways; the functions are not commutative. This approach is conceptually distinct from other object interpolation methods, which decode the interpolation of two different latent vectors. In our formulation, we visualize the outputs of an encoder that has been trained to output 2-mappings in $\mathbb{R}^3$. In addition, the bottom of Figure 4 demonstrates a smooth gradient of compositions of the reconstruction functions for two airplanes, when a higher order of mappings ($k = 4$) is used.

To further convey the expressiveness of the composition-based object interpolation, we compare it against a method that performs interpolation in the network parameter space. This latter approach resembles a common way of performing object interpolation in LVC methods: Generate latent codewords from each image, and synthesize new objects by feeding the interpolated latent vectors into the decoder. As a proxy for the latent vector interpolation used in LVC methods, we generate new objects as follows. After outputting the network parameters $\theta_A$ and $\theta_B$ for the objects $O_A$ and $O_B$, we use the interpolated parameters $\theta' = (\theta_A + \theta_B)/2$ to represent the mapping function. In Figure 7, we show that our composition-based interpolation is more capable of generating coherent new objects whose geometric features inherited from the source objects are preserved better.

## 5 CONCLUSION AND FUTURE WORK

We presented *Higher Order Function Networks* (HOF), which generate a functional representation of an object from an RGB image. The function can be represented as a small MLP with 'fast-weights', or weights that are output by an encoder network $g_\phi$ with learned, fixed weights. HOF demonstrates state-of-the-art reconstruction quality, as measured by Chamfer distance and F1 score with ground truth models, with far fewer decoder parameters than existing methods. Additionally, we extended contextual mapping methods to allow for interpolation between objects by composing the roots of their corresponding mapping functions. Another advantage of HOF is that points on the surface can be sampled directly, without expensive post-processing methods such as estimating level sets.

For future work, we would like to further improve on the parameter-efficiency of HOF, for example with versions of HOF that output only a sparse but flexible subset of the parameters of the mapping function. In addition, connections with other works investigating the properties of 'high-quality'

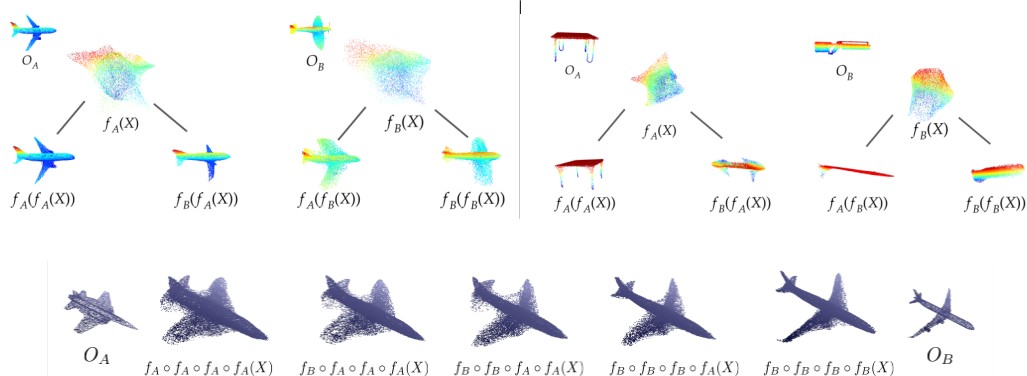

Figure 4: **Top Left.** An example of inter-class interpolation between two objects by function composition. We show the ground truth objects $O_A$ and $O_B$, a single evaluation of their respective decoding functions (giving $f_A(X)$ and $f_B(X)$), as well as the possible permutations of compositions, which makes up the leaf nodes in each tree. In $f_B(f_A(X))$, we see the wings straighten but remain narrow. In $f_A(f_B(X))$, we observe the wings broaden, but they remain angled. **Top Right.** An example of inter-class interpolation, mixing a table and a rifle. We observe what might be interpreted as a gun with legs in $f_B(f_A(X))$ and a table with a single coherent stock in $f_A(f_B(X))$. **Bottom.** An example of intra-class interpolation between two objects with $k = 4$.

neural network parameters and initializations such as HyperNetworks (Ha et al., 2016), the Lottery Ticket Hypothesis (Frankle and Carbin, 2018), and model-agnostic meta learning (Finn et al., 2017).

There are also many interesting applications of HOF in domains such as robotics. A demonstrative application in motion planning can be found in Appendix B.2.2, and Engin et al. (2020) explore extensions of HOF for multi-view reconstruction and motion planning. Using functional representations directly for example for manipulation or navigation tasks, rather than generating intermediate 3D point clouds, is also an interesting avenue of future work. We hope that the ideas presented in this paper provides a basis for future developments of efficient 3D object representations and neural network architectures.

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

## A    ADDITIONAL EXPERIMENTS

### A.1    ARCHITECTURAL VARIATIONS

We compare HOF trained on the subset of ShapeNet used in Lin et al. (2018) with several architectural variations, including using Resnet18 rather than our own encoder architecture, using a fast-weights decoder architecture with 6, rather than 3, hidden layers, and using tanh rather than relu in the fast-weights decoder. Results are summarized in Table 3. We find that HOF is competitive in all of the formulations, although using the tanh activation function in the decoder function instead of the relu shows a small degradation in performance. Future work might investigate more deeply what principles underlie the design of fast-weights architectures.

Table 3: Comparisons of class-weighted asymmetric Chamfer distances (CD(P,T) and CD(T,P)) of architectural variants of HOF-3 (with $k = 1$).

| Base (HOF-3) | ResNet18 Encoder | Deep Decoder | Tanh (Decoder only) |
|---|---|---|---|
| 1.480 / 1.014 | 1.529 / 0.989 | 1.499 / 0.998 | 1.652 / 1.066 |

### A.2    THE ROLE OF THE SAMPLING DOMAIN

In the experimental results reported in the main paper, we sample input points from the interior of the unit sphere uniformly at random. However, we might sample other topologies as input to the mapping function. In this experiment, we compare the performance of HOF when we sample from the interior of the 3D sphere (standard configuration), the surface of the 3D sphere, the interior of the 3D cube, and the interior of the 4D sphere.

Table 4: Comparisons of HOF-3 using various different canonical objects for sampling.

| 3D Sphere | 3D Sphere (Surface) | 4D Sphere | 3D Cube |
|---|---|---|---|
| 1.480 / 1.014 | 1.494 / 1.263 | 1.478 / 0.912 | 1.481 / 0.999 |

### A.3    COMPARING DIFFERENT VALUES OF K FOR COMPOSITION

Here, we compare the performance of various instances of HOF when we vary the value of $k$, or number of self-compositions. We use the HOF-1 decoder architecture (1 hidden layer with 1024 neurons) and a Resnet18 encoder architecture. This encoder architecture is different from the baseline encoder used for the results in Table 1, which explains deviations in performance from those numbers. Although performance does not vary significantly, there are some differences in reconstruction quality across values of $k$. Future work might investigate how best to take use of self-compositional or recurrent decoder architectures to maximize both computational performance as well as accuracy.

Table 5: Comparisons of Resnet18 HOF-1 with different values of $k$

| $k = 1$ | $k = 2$ | $k = 3$ | $k = 4$ |
|---|---|---|---|
| 1.635 / 1.008 | 1.574 / 1.000 | 1.601 / 1.081 | 1.655 / 0.953 |

### A.4    PROJECTION REGULARIZATION

Intuitively, we might expect that $f_\theta(X)$ would approximate the Euclidean projection function; e.g. $f_\theta(X) \approx \text{Proj}_Y(X)$. However, qualitatively, we find that this is not the case. Our mapping $f_\theta$ learns a less interpretable mapping from the canonical set $X$ to the object $O$. In order to encourage the mapping to produce a more interpretable mapping from the canonical set $X$ to the object $O$, we can regularize the transform $f_\theta$ to penalize the 'distance traveled' of points transformed by $f_\theta$. A regularization term with a small coefficient ($\lambda = 0.01$) is effective in encouraging this behavior.

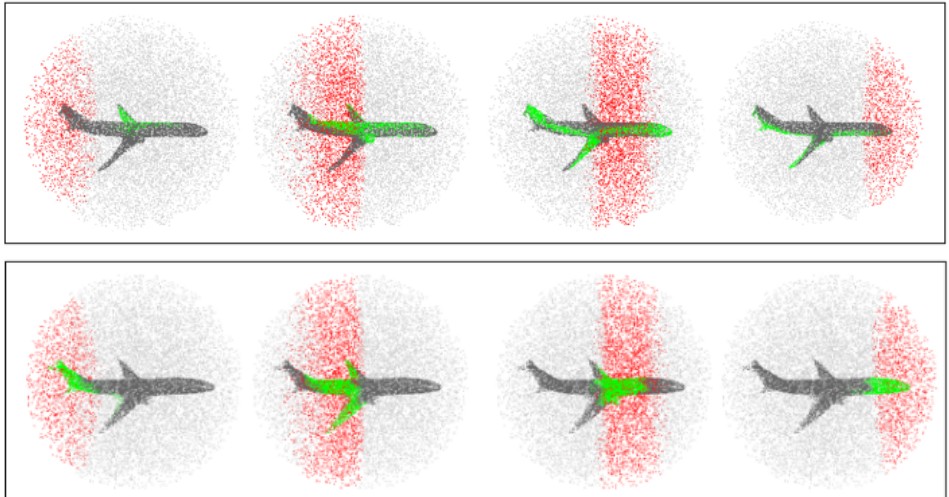

Figure 5: Slices of the sphere where the input points are sampled from and their projections in the predicted point set. Slices of the input sphere are colored red; the locations where this subset of the inputs are mapped are green. **Above**: minimizing the Chamfer distance only. **Below**: Minimizing Chamfer distance with regularization (Equation 6). In both cases, the mapping is smooth, but only with regularization is the mapping close to the intuitive projection mapping.

Making this change results in little deviation in performance, while providing a more coherent mapping. Figure 5 highlights this distinction.

This penalty for the mapping computed by $f_{\theta_I}$ for each point in the sample $\tilde{X}$ is given as

$$R(f_{\theta_I}, \tilde{X}) = \frac{1}{\tilde{X}} \sum_{\mathbf{x}_i \in \tilde{X}} ||f_{\theta_I}(\mathbf{x}_i) - \mathbf{x}_i||_2^2 \qquad (6)$$

where $\tilde{X}$ is a sample from the canonical set $X$. We might instead directly penalize the difference between $f_{\theta_I}$ and the Euclidean projection over the sampled set $\tilde{X}$ as:

$$R(f_{\theta_I}, \tilde{X}) = \frac{1}{\tilde{X}} \sum_{\mathbf{x}_i \in \tilde{X}} ||f_{\theta_I}(\mathbf{x}_i) - \mathrm{argmin}_{\mathbf{o}_i \in O}||\mathbf{o}_i - \mathbf{x}_i||_2||_2^2 \qquad (7)$$

However, we find that this regularization can be overly constraining, for example in cases where points are sampled near the boundaries of the Voronoi tesellation of the target point cloud. The formulation in Equation 6 gives the mapping greater flexibility while still giving the desired semantics.

## A.5 Collision-free Path Generation

From Chamfer distance or F1 scores alone, it is difficult to determine if one method's reconstruction quality is meaningfully different from another's. In this section, we introduce a new benchmark to evaluate the practical implications of using 3D reconstructions for collision-free path generation. We compare the reconstructions of HOF with Lin et al. (2018), which is the most competitive direct decoding method according to Table 1. This experiment is intended to give an additional perspective on what a difference in average Chamfer distance to the ground truth object means. We show that given an RGB image, we can efficiently find a near-optimal path $\hat{P}$ between two points around the bounding sphere of the object without colliding with it, and without taking a path much longer than the optimal path $P^*$, where the optimal path is defined as the shortest collision-free path between two given points. A complete definition of the experiment and its implementation are given in Section B.2.2.

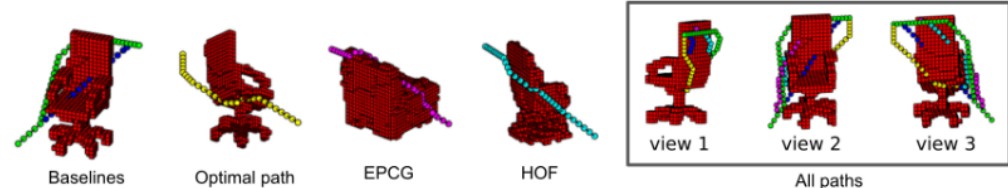

Figure 6: Collision-free path generation. The paths are color-coded as **Blue**: baseline showing $L_1$ distance, **Green**: baseline going around a bounding box around the object, **Yellow**: with GT voxels as obstacles, **Magenta**: with EPCG voxels, **Cyan**: with HOF voxels. The rightmost figure shows all paths together viewed from three different view points. Best viewed in color.

We quantify the quality of our predictions by measuring both *i*) the proportion of predicted paths $\hat{P}$ that are collision-free and *ii*) the average ratio of the length of a collision-free estimated path $\hat{P}$ and the corresponding optimal path $P^*$.

These two metrics conceptually mirror the backward and forward Chamfer distance metrics, respectively; a low collision rate corresponds to few missing structures in the reconstructed object (backward Chamfer, or surface coverage), while successful paths close to the optimal path length correspond to few extraneous structures in the reconstruction (forward Chamfer, or shape similarity).

We find that HOF provides meaningful gains over the reconstruction method recently proposed in Lin et al. (2018) in the context of path planning around the reconstructed model. HOF performs significantly better both in terms of path length as well as collision rate. However, although in Lin et al. (2018) results were reported on the reconstruction task with objects in a canonical frame, in the context of robotics, learning in a viewer-centric frame is necessary. It has been noted in Wu et al. (2018) that generalization might be easier when learning reconstruction in a viewer-oriented frame. We test this theory by training on both objects in their canonical frame as well as in the 'camera' frame. We rotate each point cloud $Y$ into its camera frame orientation using the azimuth and elevation values for each image. We rotate the point cloud about the origin, keeping the bounding box centered at (0,0,0). Trained and tested in the viewer-centric camera frame, HOF performs even better than in the canonical frame, giving Chamfer distance scores of 1.486 / 0.979 (compared with 1.534 / 1.046 for the canonical frame). The most notably improved classes in the viewer-centric evaluation are cabinets and loudspeakers; it is intuitive that these particularly 'boxy' objects might be better reconstructed in a viewer-centric frame, as their symmetric nature might make it difficult to identify their canonical frame from a single image.

Results of this comparison, as well as other ablation studies, are reported in Supplementary Tables 3 and 4. The path quality performances of the baseline metrics, EPCG Lin et al. (2018) and HOF are presented in Table 6.

Table 6: Mean values for the collision-free path generation success rate and the optimality of the output paths. Higher values indicate better performance. Details about baseline metrics *Shortest $L_1$* and *Shortest Around Bounding Box* (*SABB*) are listed in Supplementary Section B.2.2.

| Method | Shortest $L_1$ | SABB | EPCG (Lin et al., 2018) | Ours |
|---|---|---|---|---|
| Success rate | $0.341 \pm 0.35$ | $\mathbf{1.0 \pm 0.0}$ | $0.775 \pm 0.19$ | $\mathbf{0.989 \pm 0.06}$ |
| Optimality | $\mathbf{1.0 \pm 0.0}$ | $0.960 \pm 0.05$ | $0.994 \pm 0.02$ | $\mathbf{0.998 \pm 0.01}$ |

A.6 PARAMETER INTERPOLATION VS FUNCTION COMPOSITION

In order to illustrate the non-trivial mappings learned by HOF for $k > 1$, we compare the reconstructions acquired from interpolating naively between decoder parameters of two different objects and the reconstructions acquired by composing the two reconstruction functions. Results are shown in Figure 7.

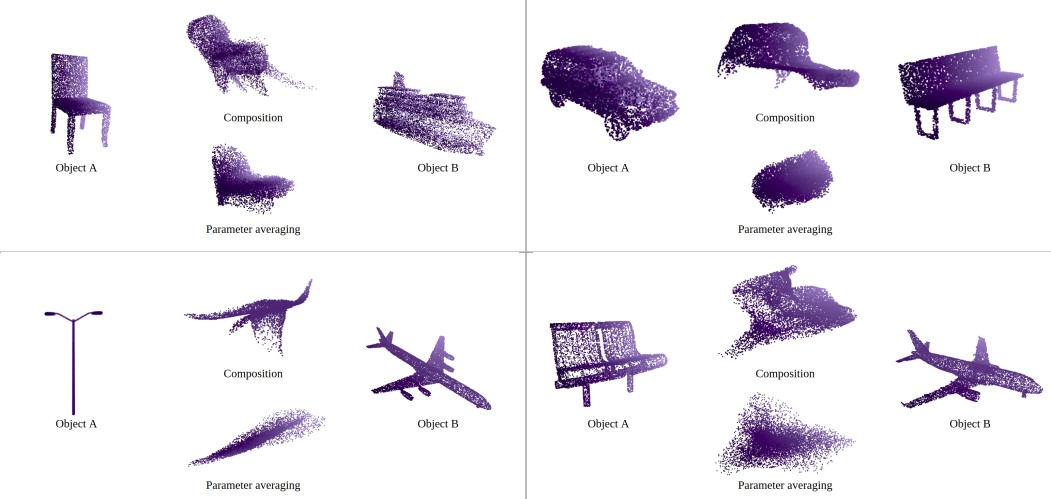

Figure 7: Inter-class interpolation between two objects using function composition and parameter interpolation. We see that the composition-based interpolation preserves some geometric features such as chair legs, car roof, and airplane wings. Direct interpolation of the network parameters fails to meaningfully capture features from the parent objects. We use $k = 4$, and $(f_B \circ f_B \circ f_A \circ f_A)(X)$ for the interpolations with composition.

### A.7 FULL CLASS BREAKDOWN FOR LVC EXPERIMENT CHAMFER DISTANCE SCORES

Table 7 shows class-wise Chamfer Distances for HOF and the baseline methods in the LVC experiment.

Table 7: Class-weighted asymmetric Chamfer distance results for our method compared to other recent methods for 3D reconstruction from images as reported in Lin et al. (2018). We use the HOF-3 architecture with $k = 1$.

| Category | 3D-R2N2 | PSG | EPCG | HOF (Ours) |
|---|---|---|---|---|
| Airplane | 2.399 / 2.391 | 1.301 / 1.488 | 1.294 / 1.541 | **0.936 / 0.723** |
| Bench | 2.323 / 2.603 | 1.814 / 1.983 | 1.757 / 1.487 | **1.288 / 0.914** |
| Cabinet | **1.420** / 2.619 | 2.463 / 2.444 | 1.814 / **1.072** | 1.764 / 1.383 |
| Car | 1.664 / 3.146 | 1.800 / 2.053 | **1.446** / 1.061 | 1.367 / **0.810** |
| Chair | 1.854 / 3.080 | 1.887 / 2.355 | 1.886 / 2.041 | **1.670 / 1.147** |
| Display | 2.088 / 2.953 | 1.919 / 2.334 | 2.142 / 1.440 | **1.765 / 1.130** |
| Lamp | 5.698 / 7.331 | 2.347 / 2.212 | 2.635 / 4.459 | **2.054 / 1.325** |
| Loudspeaker | 2.487 / 4.203 | 3.215 / 2.788 | 2.371 / 1.706 | **2.126 / 1.398** |
| Rifle | 4.193 / 2.447 | 1.316 / 1.358 | 1.289 / 1.510 | **1.066 / 0.817** |
| Sofa | 2.306 / 3.196 | 2.592 / 2.784 | 1.917 / 1.423 | **1.666 / 1.064** |
| Table | 2.128 / 3.134 | 1.874 / 2.229 | 1.689 / 1.620 | **1.377 / 0.979** |
| Telephone | 1.874 / 2.734 | 1.516 / 1.989 | 1.939 / 1.198 | **1.387 / 0.944** |
| Watercraft | 3.210 / 3.614 | 1.715 / 1.877 | 1.813 / 1.550 | **1.474 / 0.967** |
| Mean | 2.588 / 3.342 | 1.982 / 2.146 | 1.846 / 1.701 | **1.534 / 1.046** |

### A.8 FULL CLASS BREAKDOWN FOR BROAD EXPERIMENT F1 SCORES

Table 8 shows class-wise performance of HOF as well as each method compared in the broad ShapeNet comparison as reported by Tatarchenko et al. (2019).

## B  TRAINING/TESTING DATASET AND IMPLEMENTATION DETAILS

In the reconstruction experiment, Chamfer Distance scores are scaled up by 100 as in Lin et al. (2018) for easier comparison. For the numbers reported in Table 7, we use the best performance of 3d-r2n2 (5 views as reported in Lin et al. (2018)). In comparing with methods like FoldingNet Yang et al. (2018) and DeepSDF Park et al. (2019), we focus on efficiency of representation rather than reconstruction quality. The performance comparison in Figure 3 and the ablation experiment in Tables 3 attempt to compare these architectures in this way (FoldingNet is a slightly shallower version of the DeepSDF architecture; 6 rather than 8 fully-connected layers).

### B.1  DATASET

We use two different datasets for evaluation. First, in Table 1, we use the ShapeNet train/validation/test splits of a subset of the ShapeNet dataset (Chang et al., 2015) described in Yan et al. (2016). The dataset can be downloaded from `https://github.com/xcyan/nips16_PTN`. Point clouds have 100k points. Upon closer inspection, we have found that this subset includes some inconsistent/noisy labels, including:

1. Inconsistency of object interior filling (e.g. some objects are only surfaces, while some have densely sampled interiors)

2. Objects with floating text annotations that are represented in the point cloud model

3. Objects that are inconsistently small (scaled down by a factor of 5 or more compared to other similar objects)

Although these types of inconsistencies are rare, they are noteworthy. We used them as-is, but future contributions might include both 'cleaned' and 'noisy' variants of this dataset. Learning from noisy labels is an important problem but is orthogonal to 3D reconstruction.

In our second experiment, we use a broader dataset based on ShapeNet, with train and test splits taken from Tatarchenko et al. (2019). The dataset can be downloaded from `https://github.com/lmb-freiburg/what3d`.

### B.2  IMPLEMENTATION DETAILS

#### B.2.1  NETWORK ARCHITECTURE AND TRAINING

For the problem of 3D reconstruction from an RGB image, which we address here, we represent $g_\phi$ as a convolutional neural network based on the DenseNet architecture proposed in Huang et al. (2017). We call this our baseline encoder network. The baseline encoder network has 3 dense blocks (each containing 4 convolutional layers) followed by 3 fully connected layers. The schedule of feature maps is [16, 32, 64] for the dense blocks. Each fully connected layer contains 1024 neurons.

We use two variants of the mapping architecture $f_\theta$. One, which we call HOF-1, is an MLP with 1 hidden layer containing 1024 neurons. A second version, HOF-3, is an MLP with 3 hidden layers, each containing 128 hidden units. Both formulations use the ReLU activation function Glorot et al. (2011). Because $g_\phi$ and $f_\theta$ are all differentiable almost everywhere, we can train the entire system end-to-end with backpropagation. We use the Adam Optimizer with learning rate 1e-5 and batch size 1, training for 4 epochs for all experiments (1 epoch $\approx$ 725k parameter updates). Training HOF from scratch took roughly 36 hours.

#### B.2.2  PATH PLANNING EXPERIMENT

We use the class 'chair' from the dataset described in Section 4.1 in our experiments. The objects from this class have considerable variation and complexity in shape, thus they are useful for evaluating the quality of the generated paths.

Path planning is performed in a three dimensional grid environment. All the objects in our dataset fit inside the unit cube. Given the predicted point cloud of an object, we first voxelize the points by constructing an occupancy map $V = n \times n \times n$ centered at the origin of the object with voxel size $2/n$. Next, we generate start and end points as follows. We choose a unit vector $\mathbf{v}$ sampled uniformly

at random and compute $\mathbf{d} = n/2 \cdot \mathbf{v}/||\mathbf{v}||_1$. We use the end points of $\mathbf{d}$ and $-\mathbf{d}$ as the start and goal locations. For each method, we generate the paths with the A* algorithm using the voxelization of the predicted point clouds as obstacles, and the sampled start and goal positions. The movement is rectilinear in the voxel space and the distances are measured with the $L_1$ metric. In the experiments we use an occupancy grid of size $32 \times 32 \times 32$, and sample 100 start and goal location pairs per model.

In addition to the paths generated using the predictions from the EPCG Lin et al. (2018) and HOF methods, we present two other baselines (Figure 8). The first baseline *Shortest $L_1$* outputs the shortest path with the $L_1$ metric ignoring the obstacles in the scene, and the second baseline *Shortest Around Bounding Box* (*SABB*) takes the bounding box of the ground truth voxels as the environment to generate the path.

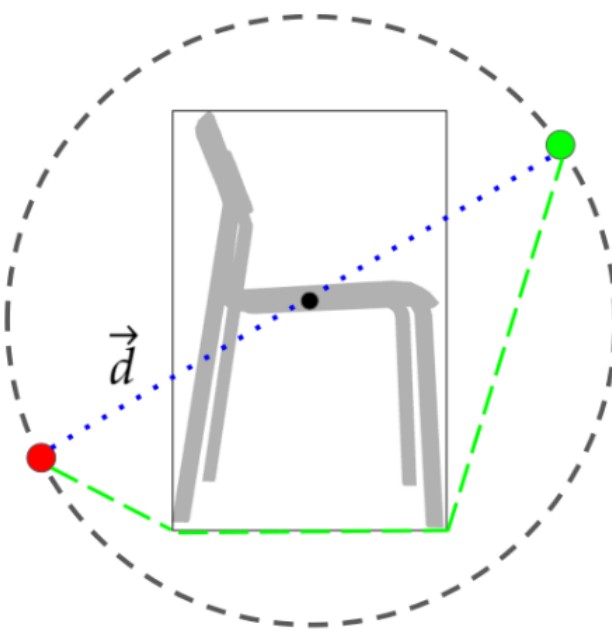

Figure 8: 2D illustration of the baseline path generation methods. The dotted path in blue is produced by *Shortest $L_1$* and the dashed path in green is by *SABB*. In our experiments, we use the rectilinear shortest path as the output of *Shortest $L_1$*.

We present the path generation results in Table 6. The baseline *Shortest $L_1$* gives the optimal solution when the path is collision-free. However, since most of the shortest paths go through the object, this baseline has a poor success rate performance. In contrast, *SABB* output paths are always collision-free as the shortest path is computed using the bounding box of the true voxelization as the obstacles in the environment. The length of the paths generated by *SABB* are longer compared to the rest of the methods since the produced paths are 'cautious' to not collide with the object. These two baselines are the best performers for the metric they are designed for, yet they suffer from the complementary metric. Our method on the other hand achieves almost optimal results in both metrics due to the good quality reconstructions.

### B.2.3    COMPUTING ENVIRONMENT

All GPU experiments were performed on NVIDIA GTX 1080 Ti GPUs. The CPU running times were computed on one of 12 cores of an Intel 7920X processor.

Table 8: F-score evaluation (@1%) on the Tatarchenko et al. (2019) dataset

| Category | AtlasNet | OGN | Matryoshka | Retrieval | Oracle NN | HOF (Ours) |
|---|---|---|---|---|---|---|
| airplane | 0.39 | 0.26 | 0.33 | 0.37 | **0.45** | 0.313 |
| ashcan | 0.18 | 0.23 | **0.26** | 0.21 | 0.24 | 0.239 |
| bag | 0.16 | 0.14 | 0.18 | 0.13 | 0.15 | **0.192** |
| basket | 0.19 | 0.16 | 0.21 | 0.15 | 0.15 | **0.217** |
| bathtub | 0.25 | 0.13 | 0.26 | 0.22 | **0.26** | 0.256 |
| bed | 0.19 | 0.12 | 0.18 | 0.15 | 0.17 | **0.195** |
| bench | 0.34 | 0.09 | 0.32 | 0.3 | 0.34 | **0.359** |
| birdhouse | 0.17 | 0.13 | **0.18** | 0.15 | 0.15 | 0.175 |
| bookshelf | 0.24 | 0.18 | 0.25 | 0.2 | 0.2 | **0.274** |
| bottle | 0.34 | 0.54 | 0.45 | 0.46 | **0.55** | 0.497 |
| bowl | 0.22 | 0.18 | 0.24 | 0.2 | 0.25 | **0.267** |
| bus | 0.35 | 0.38 | 0.41 | 0.36 | **0.44** | 0.365 |
| cabinet | 0.25 | 0.29 | **0.33** | 0.23 | 0.27 | 0.311 |
| camera | 0.13 | 0.08 | 0.12 | 0.11 | 0.12 | **0.153** |
| can | 0.23 | **0.46** | 0.44 | 0.36 | 0.44 | 0.416 |
| cap | 0.18 | 0.02 | 0.15 | 0.19 | **0.25** | 0.174 |
| car | 0.3 | 0.37 | 0.38 | 0.33 | **0.39** | 0.323 |
| cellular | 0.34 | 0.45 | 0.47 | 0.41 | **0.5** | 0.415 |
| chair | 0.25 | 0.15 | **0.27** | 0.2 | 0.23 | 0.259 |
| clock | 0.24 | 0.21 | 0.25 | 0.22 | **0.27** | 0.264 |
| dishwasher | 0.2 | 0.29 | 0.31 | 0.22 | 0.26 | **0.393** |
| display | 0.22 | 0.15 | 0.23 | 0.19 | 0.24 | **0.26** |
| earphone | 0.14 | 0.07 | 0.11 | 0.11 | 0.13 | **0.23** |
| faucet | **0.19** | 0.06 | 0.13 | 0.14 | 0.2 | 0.167 |
| file | 0.22 | 0.33 | **0.36** | 0.24 | 0.25 | 0.21 |
| guitar | 0.45 | 0.35 | 0.36 | 0.41 | **0.58** | 0.329 |
| helmet | 0.1 | 0.06 | 0.09 | 0.08 | 0.12 | **0.515** |
| jar | 0.21 | 0.22 | **0.25** | 0.19 | 0.22 | 0.114 |
| keyboard | 0.36 | 0.25 | 0.37 | 0.35 | **0.49** | 0.263 |
| knife | 0.46 | 0.26 | 0.21 | 0.37 | **0.54** | 0.504 |
| lamp | 0.26 | 0.13 | 0.2 | 0.21 | 0.27 | **0.309** |
| laptop | 0.29 | 0.21 | **0.33** | 0.26 | **0.33** | 0.284 |
| loudspeaker | 0.2 | 0.26 | **0.27** | 0.19 | 0.23 | 0.252 |
| mailbox | 0.21 | 0.2 | 0.23 | 0.2 | 0.19 | **0.292** |
| microphone | 0.23 | 0.22 | 0.19 | 0.18 | 0.21 | **0.315** |
| microwave | 0.23 | **0.36** | 0.35 | 0.22 | 0.25 | 0.279 |
| motorcycle | 0.27 | 0.12 | 0.22 | 0.24 | **0.28** | 0.27 |
| mug | 0.13 | 0.11 | 0.15 | 0.11 | **0.17** | 0.128 |
| piano | 0.17 | 0.11 | 0.16 | 0.14 | 0.17 | **0.19** |
| pillow | 0.19 | 0.14 | 0.17 | 0.18 | **0.3** | 0.227 |
| pistol | 0.29 | 0.22 | 0.23 | 0.25 | 0.3 | **0.307** |
| pot | 0.19 | 0.15 | 0.19 | 0.14 | 0.16 | **0.2** |
| printer | 0.13 | 0.11 | 0.13 | 0.11 | 0.14 | **0.17** |
| remote | 0.3 | 0.33 | 0.31 | 0.31 | 0.37 | **0.395** |
| rifle | 0.43 | 0.28 | 0.3 | 0.36 | **0.48** | 0.47 |
| rocket | 0.34 | 0.2 | 0.23 | 0.26 | 0.32 | **0.369** |
| skateboard | 0.39 | 0.11 | **0.39** | 0.35 | 0.47 | 0.384 |
| sofa | 0.24 | 0.23 | **0.27** | 0.21 | **0.27** | 0.262 |
| stove | 0.2 | 0.19 | 0.24 | 0.18 | 0.19 | **0.247** |
| table | 0.31 | 0.24 | **0.34** | 0.26 | **0.34** | 0.325 |
| telephone | 0.33 | 0.42 | **0.45** | 0.4 | 0.5 | 0.408 |
| tower | 0.24 | 0.2 | 0.25 | 0.25 | 0.25 | **0.336** |
| train | 0.34 | 0.29 | 0.3 | 0.32 | **0.38** | 0.375 |
| vessel | 0.28 | 0.19 | 0.22 | 0.23 | 0.29 | **0.299** |
| washer | 0.2 | **0.31** | **0.31** | 0.21 | 0.25 | 0.268 |

