# OpenReview forum: "Higher-Order Function Networks for Learning Composable 3D Object Representations"
_ICLR.cc/2020/Conference — Accept (Poster)_

### Official Review · AnonReviewer3 · 2019-10-21
**Official Blind Review #3**

**Rating:** 6

**Review:**

This paper presents a method for single image 3D reconstruction. It is inspired by implicit shape models, like presented in Park et al. and Mescheder et al., that given a latent code project 3D positions to signed distance, or occupancy values, respectively. However, instead of a latent vector, the proposed method directly outputs the network parameters of a second (mapping) network that displaces 3D points from a given canonical object, i.e., a unit sphere. As the second network maps 3D points to 3D points it is composable, which can be used to interpolate between different shapes. Evaluations are conducted on the standard ShapeNet dataset and the yields results close to the state-of-the-art, but using significantly less parameters.

Overall, I am in favour of accepting this paper given some clarifications and improving the evaluations.

The core contribution of the paper is to estimate the network parameters conditioned on the input (i.e., the RGB image). As noted in the related work section this is not a completely new idea (cf. Schmidhuber, Ha et al.). There are a few more references that had similar ideas and might be worth adding: Brabandere et al. "Dynamic Filter Networks", Klein et al. "A dynamic convolutional layer for short range weather prediction", Riegler et al. "Conditioned regression models for non-blind single image super-resolution", and maybe newer works along the line of Su et al. "Pixel-Adaptive Convolutional Neural Networks".

The input 3D points are sampled from a unit sphere. Does this imply any topological constraints? Is this the most suitable shape to sample from? How do you draw samples from the sphere (Similarly, how are the points sampled for the training objects)? What happens if you instead densely sample from a 3D box (similar to the implicit shape models)?

On page 4 the mapping network is described as a function that maps c-dimensional points to 3D points. What is c? Isn't it always 3, or how else is it possible to composite the mapping network?

Regarding the main evaluation: The paper follows the "standard" protocol on ShapeNet. Recently, Tatarchenko et al. showed in "What Do Single-view 3D Reconstruction Networks Learn?" shortcomings of this evaluation scheme and proposed alternatives. It would be great if this paper could follow those recommendations to get better insights in the results.
Further, I could not find what k was set to in the evaluation of Tab. 1. It did also not match any numbers in Tab. 4 of the appendix. Tab. 4 shows to some extend the influence of k, but I would like to see a more extensive evaluation. How does performance change for larger k, and what happens if k is larger at testing then on at training, etc.?

Things to improve the paper that did not impact the score:
- The tables will look a lot nicer if booktab is used in LaTeX


**Experience Assessment:**

I have published one or two papers in this area.

**Review Assessment: Checking Correctness Of Derivations And Theory:**

I carefully checked the derivations and theory.

**Review Assessment: Checking Correctness Of Experiments:**

I carefully checked the experiments.

**Review Assessment: Thoroughness In Paper Reading:**

I read the paper thoroughly.

---

> ### Author Response · Authors · 2019-11-14
> **Response to Reviewer 3**
>
> Thank you for your comments. We have performed additional experiments in order to address them, particularly training and evaluating HOF as in [1].
>
> Q1: Evaluation as in [1]
>
> In accordance with these recommendations in [1], we have trained HOF on the dataset provided by the authors of [1] and evaluated it according to the F1 metric that is proposed. We find that HOF provides competitive performance to existing methods, giving the highest average F1 score of all methods in [1]. We will report quantitative results on this benchmark in a revised PDF submitted before the end of the discussion period.
>
> Q2: Input shape
>
> We have performed new comparison experiments to address this question. Varying the input shape does affect performance; sampling the surface of the 3d sphere gives worse performance than sampling the interior of the sphere (1.369 average chamfer distance for surface of sphere versus 1.247 for interior). In addition, we find that sampling the interior of the 4D sphere, rather than the 3D sphere gives a fairly significant improvement in performance (1.195 average chamfer distance for 4D sphere vs 1.247 for 3D).
>
> Since we are learning an arbitrary mapping (rather than, for example, a projection to a manifold), the mapping domain does not explicitly induce any topological constraints such as genus. If our mapping didn't have to be continuous, it wouldn't matter what shape we sampled. However, because we use neural networks with continuous activation functions (relus) to represent the mapping, it must be continuous. And because the experiments above indicate that the mapping domain affects the quality of the reconstructions, it is possible that the sampling domain imposes topological constraints on the set of objects. It's difficult to say what the "best" shape to sample from is; however, in future work, we would like to investigate strategies for learning the best input shape to sample from.
>
> Q3: Sampling the input shape
>
> Our input 3D points are sampled from the interior of the unit sphere, rather than the surface (we have clarified this in the revised manuscript). Samples are drawn uniformly at random from within the sphere in order to avoid overfitting to a particular gridding structure.
>
> Q4: Value of $c$ for composition networks
>
> In order to apply composition, the reviewer is correct that c must equal 3 (this is the case in our composition experiments). However, if we are not composing the reconstruction function, c could be anything greater than zero. For example, c=4 in our ablation experiment in which we sample input points from the 4D sphere rather than the 3D sphere.
>
> Q5: Clarifying values of $k$
>
> $k=1$ for both HOF models in Table 1. We have edited the manuscript to clarify this point. The small difference in results between HOF-1 in Table 1 and HOF-1 ($k=1$) in Table 4 as well as HOF-3 in Table 1 and HOF-3 ($k=1$) in Table 4 is a matter of different initialization on a later training run. We have updated these tables so that the numbers are computed from the same model, rather than separate training runs.
>
> If we keep the number of compositions fixed while training and test with a larger value of k, we observe that the performance degrades significantly. On the other hand, when we use a varying number of compositions (1,2,...,k) at training time, we find that the results do generalize to higher values of k. After several additional compositions (such as k+3, k+4) however, the results start worsening similar to the trend in the fixed k evaluation.
>
> Finally, we have updated our pdf to reflect the additional related work and stylistic suggestion that you have brought to our attention. Thank you again for your feedback, and please let us know if you have any additional questions or concerns.
>
> [1] M. Tatarchenko, S. R. Richter, R. Ranftl, Z. Li, V. Koltun, and T. Brox, “What do single-view 3d reconstruction networks learn?,” in Proceedings of the IEEE Conference on Computer Vision and Pattern Recognition, pp. 3405– 3414, 2019.

---

### Official Review · AnonReviewer2 · 2019-10-24
**Official Blind Review #2**

**Rating:** 3

**Review:**

Summary:
This paper describes a contextual encoding scheme for reconstruction of 3D pointclouds from 2D images. An encoder outputs the parameters of a hierarchy of reconstruction networks that can be applied in succession to map random samples on a unit sphere to the surface of the reconstructed shape.

Strengths:
The author's model was quite novel in my opinion. Deep 2D->3D is becoming a crowded space and there are many other models that encode image inputs, and many others that perform recursive or composition-based decoding. However, the particular link here was interesting, and I appreciate the small number of parameters resulting in solid reconstruction performance. While most related work was covered well, I believe the authors could have a more up-to-date list of recent work that reconstructs triangle-mesh representations from images [A-C] (especially since several of these methods has an architecture that involves encoding and subsequent compositional refinement).

Some of the reconstructions shown in this paper are quite impressive, and the quantitative results show outperforming 2 recent methods. I did appreciate also the novel path-based evaluation of shape accuracy in the Appendix, although it would have been helpful to see more discussion of this in the main paper.

Areas for improvement:
I found that the core technical description was quite brief and would have benefited from simply more detail and space. You have argued that your method is sensible to try (cog. sci motivations), and shown that one instance works, but what can we expect in a more mathematical or general sense? Can any sizes of encoder and mapping network fit together? How does the number of mapping layers effect performance? Won't we eventually expect vanishing/exploding gradients with particular activation and can one address this in some way?

I note that recent papers in this field tend to perform significantly more extensive experimental evaluation, typically selecting a wider range of competitors and using a number of more standardized metrics including IOU, F1 score and CD and typically repeating these at a variety of resolutions or on additional datasets or category splits etc.

Decision:
Weak reject because the idea is quite interesting, but I believe a more thorough explanation and expanded experimental comparison would be of great help to ensure the community can appreciate this work.

Additional citations suggested:

[A] Pixel2Mesh: Generating 3D Mesh Models from Single RGB Images. Wang, Zhang, Li, Fu, Liu and Jiang. ECCV 2018.
[B] MeshCNN: A Network with an Edge. Hanocka, Hertz, Fish, Giryes, Fleishman and Cohen-Or. SIGGRAPH 2019.
[C] GEOMetrics: Exploiting Structure for Graph-Encoded Objects. Smith, Fujimoto, Romero and Meger. ICML 2019.

**Experience Assessment:**

I have published in this field for several years.

**Review Assessment: Checking Correctness Of Derivations And Theory:**

I carefully checked the derivations and theory.

**Review Assessment: Checking Correctness Of Experiments:**

I carefully checked the experiments.

**Review Assessment: Thoroughness In Paper Reading:**

I read the paper thoroughly.

---

> ### Author Response · Authors · 2019-11-14
> **Response to Reviewer 2**
>
> Thank you for your thoughtful review. We have been hard at work to perform additional experiments to compare with other state of the art methods on a broader dataset. We summarize the changes here and will upload a revised the manuscript with complete quantitative evaluations before the revision deadline.
>
> Q1: More extensive evaluations
>
> In response to your comments, we have trained and tested our method on the dataset provided in [1], using the F1 score rather than Chamfer Distance in accordance with the recommendations in that work. This dataset contains more than 4 times as many classes as our original dataset. We find that HOF is competitive with the performance of various state of the art methods reported in [1], showing the highest average F1 score out of all methods compared in [1]. See Section 4.1.2 for added discussion, and the Appendix Sections A7 and A8 for complete class performance breakdowns for both our original experiments as well as the new comparison with [1]. We hope this extended comparison provides a more convincing experimental evaluation of HOF.
>
> Q2: Technical description and justification
>
> In the paper, we make the observation that codeword based approaches are equivalent to learning the biases of a fixed network, whereas the fast-weights-based HOF approach learns all of the weights. Therefore, we conclude that HOF is mathematically at least as general as codeword based architectures. We further show, with experiments, that the coding provided by HOF is more efficient than codeword-based approaches in terms of number of parameters in the decoder. There is similar evidence in the literature which suggests that fast-weights based approaches can be more efficient than static networks. However at this point, similar to our paper, the evidence is empirical and a theoretical justification of this phenomenon is missing. In response to your comments, in our concluding remarks, we mention this lack of theoretical analysis and note it as an important direction for future research
>
> Q3: Architecture of encoder/mapping network
>
> In addition to experiments on a new dataset, we have performed new evaluations of variants of HOF on our original dataset to demonstrate that HOF performs competitively even when we change the encoder architecture, decoder depth, decoder activation function, or input sampling for the decoder network. For example, using Resnet18 as the encoder network gives almost identical performance in terms of average chamfer distance on our original test set. The complete quantitative results of these comparisons will be included in an updated PDF before the end of the discussion period.
>
> Q4: Number of mapping layers
>
> Our original results reported in Table 1 compare two different mapping function architectures. HOF-1 has one hidden layer with 1024 units, HOF-3 has 3 hidden layers with 128 units each. We have updated the text to clarify this distinction. In response to your comments, we have also conducted an additional experiment with a mapping network with 6 hidden layers with 128 units each; the test performance of this architecture is almost identical to that of HOF-3 (1.2485 average Chamfer distance with 6 layers compared with 1.247 average CD for 3 layers).
>
> Q5: Vanishing/exploding gradients
>
> In all of our experiments, we address the problem of vanishing/exploding gradients by dividing by the square root of the in-degree of each neuron (as in [2]). Using the same initialization in the encoder network, we find that training a mapping function with 6 hidden layers ("HOF-6") trained easily with no modifications to our training code. Another advantage of HOF over deeper, fixed decoder architectures is that it admits extremely shallow decoders, which require less careful tuning of hyperparameters such as initialization scaling and normalization compared with deeper networks. For this work, our goal was not necessarily to find the optimal architecture for the decoder, but rather to demonstrate that the usage of the higher-order function paradigm allows for a much smaller decoder architecture than LVC methods.
>
> Thank you for bringing the additional literature to our attention. We have included it in our discussion of related work in a revised manuscript. We have also updated the text to more clearly explain the path-based evaluation and its motivation.
>
> We hope that these additional experiments better demonstrate the effectiveness of HOF as a competitive, parameter-efficient 3d reconstruction paradigm.
>
> Thank you again for your feedback.
>
> [1] M. Tatarchenko, S. R. Richter, R. Ranftl, Z. Li, V. Koltun, and T. Brox, “What do single-view 3d reconstruction networks learn?,” in Proceedings of the IEEE Conference on Computer Vision and Pattern Recognition, pp. 3405– 3414, 2019.
> [2] K. He, X. Zhang, S. Ren, and J. Sun. Delving deep into rectifiers: Surpassing human-level performance on imagenet classification. In ICCV, 2015.

---

### Official Review · AnonReviewer1 · 2019-10-26
**Official Blind Review #1**

**Rating:** 6

**Review:**

This work is focused on learning 3D object representations (decoders) that can be computed more efficiently than existing methods.  The computational inefficiency of these methods is that you learn a (big) fixed decoder for all objects (all z latents), and then need to apply it individually on either each point cloud point you want to produce, or each voxel in the output (this problem exists for both the class of methods that deform a uniform distribution R^3 -> R^3 a la FoldingNet, or directly predict the 3D function R^3 -> R e.g. DeepSDF). The authors propose that the encoder directly predict the weights and biases of a decoder network that, since it is specific to the particular object being reconstructed, can be much smaller and thus much cheaper to compute.

The authors then note the fact that their method lacks a continuous latent space that allows for interpolation, as provided by existing (VAE-like) methods. They propose to solve this by learning an MLP that produces the output by recurrent application, and then composing subapplications of different networks as a type of interpolation.

-------------------

I like this work, it addresses a real problem in a number of models for 3D representation learning (similar models are also used for e.g. cryo-EM reconstruction). While the fast weights approach is not totally original, its application to this problem is novel and very well-suited to it. I was a bit surprised by just how much the decoder network could be shrunk by using fast weights.

The paper is also quite well written. I especially like how Section 2 synthesizes existing work into model categories which make it easier to think about their relationships. I also think the explanation in Sec. 3.2, while kind of obvious, is a nice way think about decoder vs. fast weights.

I like that the authors are straightforward about the deficiency of the method (i.e. that you can't interpolate in latent space). Their proposed solution of functional composition is exceedingly clever but in my opinion too impractical to really be useful. It adds extra complexity, requires you to do function composition which may be less expressive and takes more coomputation, etc. And to what end? The purpose of generative models is not to interpolate per se; the interpolation is really a sanity check that the model is capturing the underlying distribution rather than just memorizing training examples. The function composition doesn't capture that. I think the authors should just acknowledge that you can't soundly *sample* from their generative model the way e.g. VAE or GAN allows (their function composition is not a sampling method). But I think there are lots of useful things you can do without that capability, e.g. do 3D point cloud completion, go image -> structure, etc. I think this function composition angle should be deemphasized in the title/abstract, but I think the paper stands  reasonably on its own without that.

Nits:
- In Figure 2 it's pretty hard to see the differences between the methods. What exactly is being visualized here? DeepSDF shold be visualizing surface normals vs. HOF which is point clouds, right?
- For predicting a deformation R^3 -> R^3 function composition sort of makes sense, but how generalizable is this approach e.g. to directly predicting a function R^3 -> R (a la DeepSDF)? I think there are ways this function composition approach could generalize, e.g. using skip connections and layer dropout (which encourages layers to be composable).


**Experience Assessment:**

I have read many papers in this area.

**Review Assessment: Checking Correctness Of Derivations And Theory:**

I carefully checked the derivations and theory.

**Review Assessment: Checking Correctness Of Experiments:**

I assessed the sensibility of the experiments.

**Review Assessment: Thoroughness In Paper Reading:**

I read the paper at least twice and used my best judgement in assessing the paper.

---

> ### Author Response · Authors · 2019-11-14
> **Response to Reviewer 1**
>
> Thank you for your review and comments. We’ve made a number of additions and improvements to address them in the updated version of the paper, which we will submit before the end of the discussion period.
>
> First, we have performed a new set of experiments on the larger dataset in [1]. HOF shows greater average reconstruction accuracy than the methods compared in [1]. Second, we also perform ablation experiments to demonstrate that HOF performs competitively even when we vary the encoder architecture, decoder depth, decoder activation function, or input sampling for the decoder network. For example, using Resnet18 as the encoder architecture or using a decoder network with twice as many hidden layers showed nearly identical performance in terms of average Chamfer distance on the test set. The complete quantitative results will be included in an updated PDF before the end of the discussion period.
>
> "The purpose of generative models is not to interpolate per se; the interpolation is really a sanity check that the model is capturing the underlying distribution rather than just memorizing training examples. The function composition doesn't capture that. I think the authors should just acknowledge that you can't soundly *sample* from their generative model the way e.g. VAE or GAN allows (their function composition is not a sampling method). But I think there are lots of useful things you can do without that capability, e.g. do 3D point cloud completion, go image -> structure, etc. I think this function composition angle should be deemphasized in the title/abstract, but I think the paper stands  reasonably on its own without that."
>
> We agree that the current formulation of composition is not equivalent to a generative model. In our work, function composition primarily serves the purpose of demonstrating that the model learns a meaningful subspace of objects (rather than memorizing the training set, as you mentioned). We have revised the abstract to clarify this point.
>
> "In Figure 2 it's pretty hard to see the differences between the methods. What exactly is being visualized here? DeepSDF shold be visualizing surface normals vs. HOF which is point clouds, right?"
>
> We have clarified in the manuscript that our comparisons are between architectures, rather than training objectives/output representations. Thus our DeepSDF, FoldingNet, and HOF architectures all output point clouds, which can be compared directly.
>
> "For predicting a deformation R^3 -> R^3 function composition sort of makes sense, but how generalizable is this approach e.g. to directly predicting a function R^3 -> R (a la DeepSDF)? I think there are ways this function composition approach could generalize, e.g. using skip connections and layer dropout (which encourages layers to be composable)."
>
> Additional techniques for promoting learning of composable representations such as skip connections and layer dropout are an exciting direction for future research. One way function composition might allow for R^3 -> R mappings by composing a mapping from R^3 -> R^3 and taking the only first dimension of each element in the final output.
>
> Thank you again for your feedback.
>
> [1] M. Tatarchenko, S. R. Richter, R. Ranftl, Z. Li, V. Koltun, and T. Brox, “What do single-view 3d reconstruction networks learn?,” in Proceedings of the IEEE Conference on Computer Vision and Pattern Recognition, pp. 3405– 3414, 2019.

---

### Author Response · Authors · 2019-11-15
**Summary of Revisions**

We thank the reviewers for their insightful suggestions and comments, as well as their appreciation of the novelty of our method HOF. We feel that we were able to address all of their concerns in our followup experiments. Most of all, the reviewers requested additional experiments comparing our method with other baselines. To address their comments, we trained and evaluated HOF on the dataset from [1], using the F1 metric suggested by the authors of [1] and requested by the reviewers. We report results from all 55 classes in the appendix of the revised paper. HOF surpasses the prior methods reported in [1] in terms of average F1 score over the testing set and performs competitively with the "Oracle" baseline in [1]. Excluding the Oracle, HOF gives the highest accuracy of all methods on 28 of the 55 classes. This new experiment can be found in Section 4.1.2 of the revised PDF, with complete class breakdowns located in the revised Appendix A.8.

We also conducted new comparisons with various canonical objects (2D sheet, sphere surface, complete sphere with interior, high-dimensional cube) to explore the role of the sampling domain. Finally, we demonstrated that the performance of HOF is relatively unaffected if the encoder or decoder architecture is varied. The results of these new comparisons are included in the revised Appendix. We also incorporated new related work noted by the reviewers as well as the reviewers' stylistic suggestions. Details are provided in the individual responses below.

We thank the reviewers again for their time and valuable comments. These additional experiments have established a clearer picture of our contributions. We hope that the reviewers will be satisfied with the revised version of the paper!


[1] M. Tatarchenko, S. R. Richter, R. Ranftl, Z. Li, V. Koltun, and T. Brox, “What do single-view 3d reconstruction networks learn?,” in Proceedings of the IEEE Conference on Computer Vision and Pattern Recognition, pp. 3405– 3414, 2019.

---

### Decision · Program_Chairs · 2019-12-19

**Decision:**

Accept (Poster)

**Comment:**

The submission presents an approach to single-view 3D reconstruction. The approach is quite creative and involves predicting the weights of a network that is then applied to a point set. The presentation is good. The experimental protocol is well-informed and the results are convincing. The reviewers' concerns have largely been addressed by the authors' responses and the revision. In particular, R2, who gave a "3", posted "I would now advise to raise my score (3 previously) to a be in line with the 6: Weak Accept given by the other reviewers." This means that all three reviewers recommend accepting the paper. The AC agrees.